# *TP53* Pathogenic Variants in Early-Onset Breast Cancer Patients Fulfilling Hereditary Breast and Ovary Cancer and Li-Fraumeni-like Syndromes

**DOI:** 10.3390/biom12050640

**Published:** 2022-04-27

**Authors:** Paula Francinete Faustino da Silva, Rebeca Mota Goveia, Thaís Bomfim Teixeira, Bruno Faulin Gamba, Aliny Pereira de Lima, Sílvia Regina Rogatto, Elisângela de Paula Silveira-Lacerda

**Affiliations:** 1Laboratory of Molecular Genetics and Cytogenetics, Institute of Biological Sciences I (ICB I), Federal University of Goiás, Goiânia 74690900, Brazil; paulabiomedicina02@hotmail.com (P.F.F.d.S.); rebecamg2013@gmail.com (R.M.G.); gamba.bf@hotmail.com (B.F.G.); alinypereiralima@gmail.com (A.P.d.L.); 2Clinical of Hospital (HC), Federal University of Goiás (UFG), Goiânia 74690900, Brazil; thaisbteixeira@gmail.com; 3Department of Clinical Genetics, University Hospital of Southern Denmark, 7100 Vejle, Denmark; silvia.regina.rogatto@rsyd.dk; 4Institute of Regional Health Research, University of Southern Denmark, 5230 Odense, Denmark

**Keywords:** *TP53*, breast cancer, Li-Fraumeni syndrome, cancer predisposition

## Abstract

*TP53* gene mutation is the most common genetic alteration in human malignant tumors and is mainly responsible for Li-Fraumeni syndrome. Among the several cancers related to this syndrome, breast cancer (BC) is the most common. The *TP53* p.R337H germline pathogenic variant is highly prevalent in Brazil’s South and Southeast regions, accounting for 0.3% of the general population. We investigated the prevalence of *TP53* germline pathogenic variants in a cohort of 83 BC patients from the Midwest Brazilian region. All patients met the clinical criteria for hereditary breast and ovarian cancer syndrome (HBOC) and were negative for *BRCA1* and *BRCA2* mutations. Moreover, 40 index patients fulfilled HBOC and the Li-Fraumeni-like (LFL) syndromes criteria. The samples were tested using next generation sequencing for *TP53*. Three patients harbored *TP53* missense pathogenic variants (p.Arg248Gln, p.Arg337His, and p.Arg337Cys), confirmed by Sanger sequencing. One (1.2%) patient showed a large *TP53* deletion (exons 2–11), which was also confirmed. The p.R337H variant was detected in only one patient. In conclusion, four (4.8%) early-onset breast cancer patients fulfilling the HBOC and LFL syndromes presented *TP53* pathogenic variants, confirming the relevance of genetic tests in this group of patients. In contrast to other Brazilian regions, *TP53* p.R337H variant appeared with low prevalence.

## 1. Introduction

Germline pathogenic variants of the *TP53* gene have been associated with a high risk of developing a particular spectrum of common tumors at early onset. Li-Fraumeni syndrome (LFS) is a rare hereditary cancer predisposition syndrome caused by *TP53* pathogenic variants [1]. The spectrum of tumor types associated with LFS includes breast cancer, adrenocortical cancer, soft tissue sarcomas, osteosarcomas, and central nervous system tumors, among others [1,2]. The clinical diagnosis of LFS is defined by the presence of an individual diagnosed with sarcoma before the age of 45 years, a first-degree relative with cancer before age 45, and a second relative, first or second degree, diagnosed with cancer before age 45 or sarcoma at any age [3]. Since its description, the clinical characterization of the LFS has changed due to observed phenotypic variations. Li-Fraumeni-like syndrome (LFL), called non-classical or variants of LFS, shares many characteristics of classical Li-Fraumeni syndrome, only less restricted. LFL proband shows the LFS tumor spectrum before age 45, plus a first- or second-degree relative who also had an LFS cancer spectrum before age 60. LFL and LFS significantly increase the chances of carriers of the *TP53* mutation developing multiple cancers since childhood [4]. 

The first study of *TP53* variants in patients with clinical criteria of LFS was described by Malkin et al. [5]. The authors reported germline variants at exons 5–8 of the *TP53* gene in individuals from five families. It is estimated that 70% of LFS patients are associated with *TP53* pathogenic variants [5,6,7]. In 22 Spanish families with LFS, Lovet et al. described that 93% of *TP53* mutation carriers develop at least one cancer type before age 35 [8]. 

Although LFS is considered a rare condition, several studies reported a high incidence in Brazil. Ribeiro et al. [9] identified a specific germline variant in the oligomerization domain of the *TP53* gene at the codon 337 (c.1010 G > A) in 35 of 37 (94.59%) children with adrenocortical carcinomas and no family history of cancer. In 2006, a study conducted on 30,098 Brazilian newborns from southern and southeastern regions found a high prevalence of carriers of p.R337H variant (0.3%), which is higher than that described worldwide (1/5000) [10]. A subsequent study evaluated 45 patients with LFL/LFS and found the same *TP53* variant in six patients (13.33%) [11].

The p.R337H variant was also described in 2.5% (3/120) of breast cancer patients fulfilling the criteria for HBOC and 8.6% (70/815) of women diagnosed with breast cancer at or before age 45 or at age 55 or older from south and southern regions [12]. Recently, Mathias et al. [13] reported that *TP53* p.R337H variant contributed to 2.4% of 805 sporadic breast cancer patients with no family cancer history from South Brazil. Studies of several populations have observed the prevalence of early-onset breast cancer in women with pathogenic germline *TP53* variants [13,14]. About 5–8% of women with breast cancer under 30 years old have a germline pathogenic *TP53* variant [15]. Women who carry *TP53* germline pathogenic variants have a high risk of breast cancer of up to 85% at age 60 [14,16].

The high prevalence of p.R337H in Brazilian families raised the question of a possible founding effect. Based on this hypothesis, Pinto et al. [17] analyzed two *TP53* intragenic polymorphic markers in 22 cases and 60 controls and suggested that it was very likely that the p.R337H germline variant has arisen from a common origin. Garritano et al. [18] used a panel of SNPs spanning the entire *TP53* gene and concluded that this rare haplotype would have an extremely low probability of arising independently (1/100,000,000), reinforcing the existence of a founding effect. Since Brazil is a highly populous country worldwide, pathogenic variants may be present in thousands of individuals, explaining the high frequency of many tumor types [19].

Considering the wide distribution of germline pathogenic variants in *TP53* in Brazilian individuals and the association with early-onset breast cancer, we investigated the presence of *TP53* variants in a cohort of breast cancer patients from the unexplored central region of Brazil.

## 2. Materials and Methods

### 2.1. Patients

A total of 83 index patients and 217 family members were included in this study. The index patients were recruited at the Clinical Genetics Ambulatory of the HC/UFG Clinical Hospital, Brazil, from 2017 to 2019. All patients provided a free signed and informed consent form following the Declaration of Helsinki and received genetic counseling. This study was approved by the Research Ethics Committee (CEP) of the Federal University of Goiás, Brazil (CEP-CONEP-CAAE: 50626315.6.00 00.5078/# 1791010). A peripheral blood sample was collected, and a questionnaire with socio-demographic data was applied. 

We included patients who met the clinical criteria for hereditary breast and ovarian cancer syndrome (HBOC) based on the NCCN Clinical Practice Guidelines on Oncology (NCCN, 2021) [20]. Patients that fulfilled the criteria for HBOC were previously tested with no detectable *BRCA1* or *BRCA2* germline variants. After collecting information about the family history of cancer, 83 index patients had a total of 217 family members with cancer. Each patient was classified according to the Chompret criteria for Li Fraumeni syndrome, as described by Bougeard et al. [21]. We analyzed *TP53* variants in 40 patients (Group 1) fulfilling the Chompret criteria and 43 patients (Group 2) who did not meet the criteria for LFS or LFL. However, all families had at least one family member with a typical LFS tumor and several other relatives with cancer (e.g., bowel, liver, prostate, and stomach cancer). Patient pedigrees were constructed using the online program Genealogy.com. A risk assessment was performed according to the criteria for clinical diagnosis of Li-Fraumeni syndrome, as described by Bougeard et al. [21].

### 2.2. Genomic DNA Isolation and Sequencing 

The DNA extraction from peripheral blood samples was performed using the PureLink Genomic DNA Mini Kit (Invitrogen, Carlsbad, CA, USA). DNA integrity and quantification were evaluated using 1.5% agarose gel electrophoresis and Qubit fluorimeter (Invitrogen, Carlsbad, CA, USA), respectively. Libraries preparation and sequencing were performed by Sophia Genetics (Saint-Sulpice, Switzerland). Libraries were prepared using the CE-IVD Sophia HCS v1.1 kit (Sophia Genetics SA HQ, Saint-Sulpice, Switzerland). Next generation sequencing (NGS) of all coding regions and intron-exon junctions of the *TP53* was carried out using the Illumina MiSeq DX platform according to Illumina (Illumina, San Diego, CA, USA) and Sophia Genetics protocols. The sequences obtained were aligned to the hg19 reference genome. Variant calling and data sequencing analysis were performed with the Sophia-DDM-V4 software (Sophia Genetics, Saint-Sulpice, Switzerland). Genetic variant annotations were compared with the literature and open-source bioinformatics tools such as ClinVar, [22] IARC *TP53* Database, [23] ABraOM, [24] 1000 Genomes Project, [25] ExAC, [26] dbSNP [27] and The Genome Aggregation Database [28]. The variants were classified as pathogenic, likely pathogenic, variants with uncertain significance, benign, and likely benign, according to the American College of Medical Genetics and Genomics. We focused on pathogenic variants.

### 2.3. Variants Confirmation by Sanger Sequencing and MLPA (Multiplex Ligation-Dependent Probe Amplification)

The *TP53* pathogenic variants were confirmed by Sanger sequencing or MLPA. The 447 bp fragment corresponding to exon 10 of the *TP53* gene was amplified by PCR, as described by Custódio et al. [29]. The amplified product was purified using Illustra ExoProStar 1-Step (GE Healthcare, Chicago, Illinois, USA), following the manufacturer’s recommendations. Next, bidirectional sequencing (Sanger method) was performed using the BigDye Terminator Cycle Sequencing v1.1 kit on the ABI 3130xL platform (Applied Biosystems, Foster City, CA, USA). After obtaining the sequences alignment using BioEdit (version X), the chromatograms were visualized and analyzed with the Chromas software 2.6.6 (Technenlysium Ltd., South Brisbane, Australia).

A large *TP53* deletion found in one case was investigated using the SALSA MLPA P056 *TP53* kit (MRC-Holland, Amsterdam, The Netherlands, lote: C1-0215), following the manufacturer’s instructions. The Coffalyser Net software (MRC-Holland, Amsterdam, The Netherlands) was used for data analyses.

### 2.4. Statistical Analysis

R Studio software (http://www.rstudio.com (accessed on 1 September 2020)) was used to calculate the mean, standard deviation, *T*-test, and ANOVA, with a 95% confidence interval (CI). The p-values of less than 0.05 were considered statistically significant.

## 3. Results

Our cohort of 83 patients that fulfilled the clinical criteria for HBOC had 217 family members with cancer. Forty of 83 index patients also met the Chompret criteria for Li Fraumeni syndrome. Both index patients (81 of 83) and their relatives (74.1%) were predominantly of female gender (161 of 217). A total of 17 index patients (20.4%) developed cancer before the age of 31. The mean age of diagnosis of the first tumor was 37.7 years (Table 1). Table 1 summarizes the clinical features of our HBOC patients. 

Breast cancer was frequently found in relatives of our index patients (Group 1: 17.5% and Group 2: 10.5%) (Figure 1A). The second most common tumor type in family members of patients from Group 1 was bowel cancer (4.6%), followed by prostate cancer (4.1%) and head and neck cancer (3.2%). Uterus cancer (4.1%) was the second most common tumor type in relatives from the index patients of Group 2, and the third was prostate cancer (3.2%) (Figure 1A). 

Four female index patients (Group 1) were carriers of *TP53* pathogenic germline variants in heterozygosis, three missense, and one deletion. Sanger sequencing confirmed the variants c.743G > A (p.Arg248Gln), c.1010G > A (p.Arg337His), and c.1009C > T (p.Arg337Cys). The deleterious variant E2-11 is a virtually complete deletion of the *TP53* gene (exons 2–11), which was confirmed by MLPA. The pedigrees of four index patients with *TP53* variants are shown in Figure 1B.

## 4. Discussion

Li-Fraumeni syndrome was previously estimated in 1% (3 of 300) of women with breast cancer fulfilling the criteria for hereditary cancer predisposition syndrome [30,31]. Carriers of the *TP53* mutation have an increased risk of developing breast cancer at early-onset compared to the general population [30,32]. In addition, it was suggested that 3.8 to 7.7% of women with breast cancer aged 30 years or less harbored *TP53* germline pathogenic variants [33,34]. 

Herein, we described four young patients (22–24 years old) (4.8%) with germline *TP53* pathogenic variants. These four patients have a family history consistent with LFL and HBOC syndromes (Group 1). The indication of simultaneous genetic testing for *BRCA1*, *BRCA2,* and *TP53* was recommended for women diagnosed with breast cancer at early onset who have a family history of cancer associated with LFS [35,36]. 

Also, current guidelines for Li-Fraumeni syndrome (2015 version of the Chompret criteria) recommend that *TP53* genetic testing be considered for women diagnosed with breast cancer before the age of 31 years [21]. In our cohort, the remaining 36 patients fulfilling the clinical criteria for HBOC and LFL syndromes were negative for *BRCA1*, *BRCA2,* and *TP53* pathogenic variants. Whole exome sequencing and copy number alterations analysis are alternative strategies to be applied to these patients.

One index patient at the age of 24 presented a deep deletion of the *TP53* gene involving the exons 2 to 11. Germline deletion of the entire exon 10 of *TP53* was reported by Plummer et al. [37] in a family with LFS. Interestingly, a complete heterozygous deletion of the *TP53* gene (45 kb) was reported in a family fulfilling the criteria of LFS and HBOC [23]. In a Spanish family, Lovet et al. [8] reported an in-frame deletion of *TP53* c.437_445del located at the DNA-binding domain (exon 5), resulting in loss of function of the protein. Large intragenic deletions of *TP53* have also been reported in the IARC *TP53* database [38] in 8% of cases. Although large intragenic deletions of the *TP53* gene are not a frequent cause of hereditary cancer syndromes, strategies to explore these events should be taken into consideration. 

Inactivation of the tumor suppressor *TP53* by missense mutations is the most frequent genetic alteration in human cancers. These missense variants disrupt the ability of p53 to bind to DNA and consequently interfere in the transactivation of downstream genes [39]. We found three missense pathogenic variants in heterozygosis. 

One of 83 index patients with HBOC and LFS (27 years old) presented the *TP53* p.R337H variant in heterozygous (1.2%). A Brazilian study in breast cancer patients identified 70 index cases with the p.R337H variant, of which 68 (97%) had a heterozygous state [12]. Breast cancer was described in 28.6% of families with the p.R337H variant [11]. This variant was associated with tumor aggressiveness, low survival rates, and poor prognosis [40]. Studies from South and Southeast Brazil have reported an incidence ranging from 2.5 to 8.5% of LFS, mainly due to the *TP53* p.R337H variant [11,12,41,42,43]. In the Northeast region of Brazil, a low frequency of this variant (0.9%) was reported in breast cancer patients [44]. Brazilian population studies using haplotypes analysis reinforce the hypothesis that the *TP53* p.R337H variant is a founder mutation that may have arisen between São Paulo (Southeast) and Porto Alegre (South) [12,17,18]. Apparently, an immigrant from Portugal was carrier of this mutation. As he worked as a drover on the route between the South and Southeast regions of Brazil, he would have left descendants carriers of this variant, which explains its high prevalence in these regions [45]. We reported for the first time the presence of the p.R337H variant in one HBOC/LFS patient from the Midwest region of Brazil. Our findings showing a low prevalence (1.2%) of the p.R337H variant reinforces the hypothesis that the founder haplotype emerged in the South region and passed through the southeast Brazilian regions, which is highly prevalent.

We also described two HBOC/LFL patients with *TP53* missense pathogenic variants: c.1009C > T (p.Arg337Cys) and c.743G > A (p.Arg248Gln). The *TP53* c.1009C > T (p.Arg337Cys) and c.743G > A (p.Arg248Gln) variants were reported by Li et al. [46] in non-BRCA1/2 women with familial breast cancer. Jouali et al. [47] reported that seven of 39 Moroccan patients with triple negative breast cancer harbored c.743G > A (p.Arg248Gln) variant. 

Our patients showing *TP53* pathogenic variants were diagnosed with breast cancer before the age of 27. According to Melhem-Bertrandt et al. [48], women with breast cancer associated with Li-Fraumeni syndrome tend to develop cancer at a very young age (20 to 30 years). These women diagnosed with breast cancer under the age of 30 generally do not have a significant family history of cancer, and 3–8% of them have a pathogenic variant of the *TP53* gene [15,49]. Therefore, it is important to prescribe genetic testing for young patients even if they do not have a strong family history of cancer. 

## 5. Conclusions

We describe a low frequency (4.8%) of the *TP53* germline pathogenic variants in the Midwest region of Brazil in breast cancer patients. In addition, the most frequent Brazilian variant p.R337H found in the South and Southeast regions (2.5 to 8.5%) was detected in only one patient (1.2%). It is noteworthy that all four carriers of the *TP53* variants have the disease at early-onset and fulfill the HBOC and LFL syndromes criteria. Therefore, in addition to screening tests for *BRCA1* and *BRCA2*, young breast cancer patients should also be investigated for *TP53* variants. 

## Figures and Tables

**Figure 1 biomolecules-12-00640-f001:**
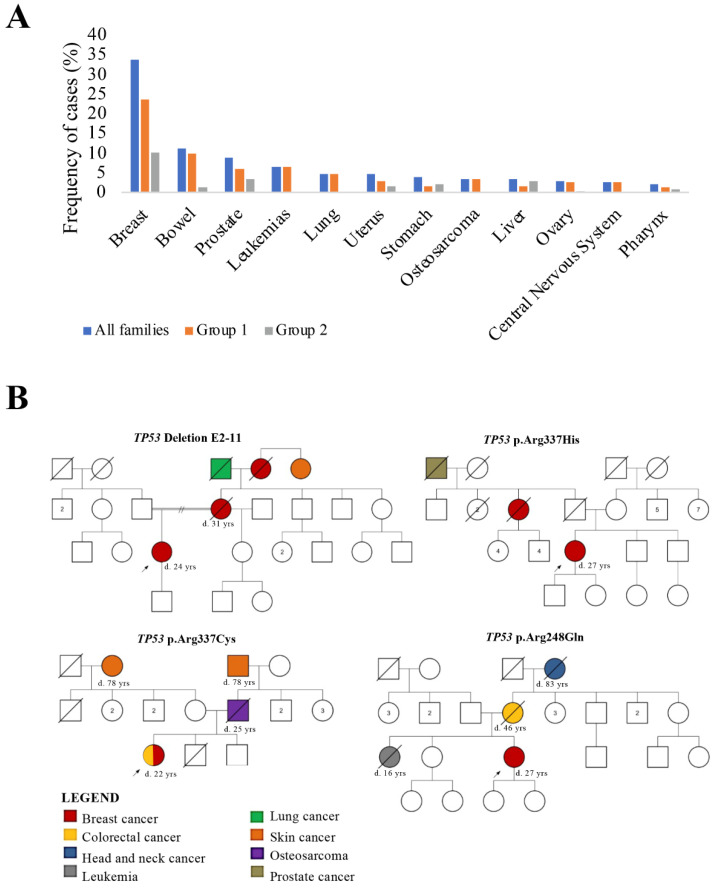
(**A**) Tumor types reported breast cancer patients and their family members. (**B**) Pedigrees of the index patients with germline *TP53* pathogenic variants. The age at diagnosis is indicated by (d.).

**Table 1 biomolecules-12-00640-t001:** Clinical characterization of breast cancer patients and family history of cancer.

	Breast Cancer Patients (NCCN, 2021)	Family Members with Cancer
Patients (N = 378)	83	217
Clinical Criteria		
Group 1: Chompret Criteria- LFL	40 (48.1%)	119 (54.8%)
Group 2: no criteria for LFS/LFL	43 (51.8%)	98 (45.1%)
Family members with cancer breast		61 (28.1%)
Group 1		38 (17.5%)
Group 2		23 (10.5%)
Gender		
Male	2 (2.4%)	56 (25.8%)
Female	81 (97.5%)	161 (74.1%)
Mean age at diagnosis	37.7 (22–64)	NA
Grupo 1	33.9 (22–57)	NA
Grupo 2	41.5 (32–64)	
<31 years old	17 (20.4%)	NA
Patients with multiple primary tumors	7	5

LFL: Li-Fraumeni-like syndrome; LFS: Li-Fraumeni syndrome; NA: not available.

## Data Availability

All data presented in this study are available in the Table 1.

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
