# Peer review of "TP53 Pathogenic Variants in Early-Onset Breast Cancer Patients Fulfilling Hereditary Breast and Ovary Cancer and Li-Fraumeni-like Syndromes"

_biomolecules, 2022, doi:10.3390/biom12050640_

Round 1
Reviewer 1 Report
This is a study focusing on TP53 testing. Based on the rarity of TP53 variants, to my view all relevant reports merit at least condideration for publication.
I am afraid that the paper is not clearly written and requires extensive editing by a native speaker.
The abstract is confusing. Try to expain simply which is your patient cohort and the selection criteria. Why chose patients for TP53 implementing both NCCN and Chomret criteria?
-Why isn't there a mutation rate on the abstract?
-Try to stick to HGVS rules for nomenclature
-Damaging variants vs pathogenic variants
-What is the meaning of lines 37-40? What is the meaning of LFL? Explain
-''It is estimated that 70% of LFS patients are associated with TP53 damaging variants'' This statement, if correct, is not supported by refs 4-6.
-''Carriers of these variants have a risk of up to 93% of developing cancer usually diagnosed before 35 years old'' This statement, if correct, is not supported by ref7.
-Consider rephrazing 3rd paragraph of introduction to make your points clear.
-Results: referring on 217 family members that were never listed on methods.
-Most of the 1st paragraph of results goes on methods.
-Your figure is not showing appropriately
-State prevalence
-Keep your discussion short and clear
Author Response
OVERALL RESPONSE
We would like to thank the reviewers for their positive comments and suggestions to improve our manuscript. All alterations included in the manuscript are highlighted in blue.
REVIEWER 01
- This is a study focusing on TP53 testing. Based on the rarity of TP53 variants, to my view all relevant reports merit at least consideration for publication. I am afraid that the paper is not clearly written and requires extensive editing by a native speaker.
Response: As recommended, the manuscript was reviewed and proofread by a native English speaker.
- Try to explain simply which is your patient cohort and the selection criteria. The abstract is confusing.
Response: We included patients that met the hereditary breast and ovarian cancer syndrome (HBOC) criteria and showed no mutations of BRCA1 and BRCA2, the main genes associated with this phenotype. The Abstract section described our sample selection criteria, as suggested (lines 18-20).
- Why chose patients for TP53 implementing both NCCN and Chompret criteria?
Response: We used the NCCN guidelines to identify patients who met the clinical criteria
for hereditary breast and ovarian cancer syndrome. A set of our patients also fulfilled the clinical criteria for Li Fraumeni syndrome (LFL). The Chompret criteria for diagnosing Li-Fraumeni Syndrome were proposed to identify affected families beyond the classic criteria adopted for Li-Fraumeni Syndrome. The overlapping of clinical criteria used for identifying a hereditary cancer syndrome is described in many cases (Stanislaw C, Xue Y, Wilcox WR. Genetic evaluation and testing for hereditary forms of cancer in the era of next-generation sequencing. Cancer Biol Med. 2016;13(1):55-67. doi:10.28092/j.issn.2095-3941.2016.0002).
- Why isn't there a mutation rate on the abstract?
Response: The mutation rate (4.8%) was included in the Abstract (page 1, line 25).
- Try to stick to HGVS rules for nomenclature
Response: The variant sequence nomenclature has been revised and followed the HGVS recommendations.
- Damaging variants vs pathogenic variants
Response: The term damaging variants were replaced by pathogenic variants across the entire manuscript.
- What is the meaning of lines 37-40? What is the meaning of LFL? Explain
Response: We included the definition of Li-Fraumeni like syndrome in detail, as recommended (lines 37 – 46).
- It is estimated that 70% of LFS patients are associated with TP53 damaging variants'' This statement, if correct, is not supported by refs 4-6.
Response: This statement is correct (for review, doi:10.1177/1947601911413466). However, we replaced some references (6, 7, and 8) to support this information in the Introduction section (lines 49-51).
9.''Carriers of these variants have a risk of up to 93% of developing cancer usually diagnosed before 35 years old'' This statement, if correct, is not supported by ref7.
Response: In 22 Spanish families with Li Fraumeni syndrome, Lovet et al. (Fam Cancer 2017 Oct;16(4):567-575. doi: 10.1007/s10689-017-9990-0) described that 93% of TP53 mutation carriers developed at least one malignancy (mainly breast cancer and sarcomas). To clarify, we modified the sentence (lines 50-51).
- Consider rephrasing 3rd paragraph of introduction to make your points clear.
Response: The 3rd paragraph was reformulated, as recommended.
- Results: referring on 217 family members that were never listed on methods.
Response: The reviewer is correct. We included the number of family members (affected by cancer) of our index patients in the Material and Methods section, Patients (line 84; lines 95-96).
- Most of the 1st paragraph of results goes on methods.
Response: We reformulated the first paragraph (page 3, lines 141 – 142). However, in our opinion, the information presented is essential to contextualize the results presented in this section.
- Your figure is not showing appropriately
Response: The figure was modified and presented in high resolution (600dpi).
- State prevalence
Response: Unfortunately, the State prevalence of HBOC/LFS cancer syndromes is unknown. To our knowledge, this is the first study describing the mutational status of TP53 in the Midwest region of Brazil (lines 217-219). The prevalence is presented on page 5 (line 179). The Discussion (lines 203-204) and the Conclusion (line 236) sections were properly altered.
15.Keep your discussion short and clear
Response: As recommended, we revised the Discussion section.

Reviewer 2 Report
General comments
Silva et al. investigated TP53 gene mutations in 83 Breast Cancer (BC) patients and their 217 family members in mid-west Brazilian region. They identified 4 at the age of 22- 24 out of 40 Chompret criteria Li Fraumeni syndrome patients with germline TP53 pathogenic variants, including the founder mutation p.R337H, while the remainder 36 patients were tested negative for BRCA1, BRCA2 and TP53 pathogenic variants. The study adds useful information for TP53-LFS/LFL-HBOC in the midewest Brazil region.
Concerns:
- It will be interesting to provide further info for why p.R337His high in other region but low in midwestern region? Any population history info, dating info etc. to enhance the explanation?
- There are multiple grammatical errors in table and poor quality of figures that Figure 1 is largely messed up for patient information
- Line 136 – The text written 161 out of 217 are female, while in table 1, 149 out of 217 are females.
- In consistent reference format with main manuscript.
Author Response
OVERALL RESPONSE
We would like to thank the reviewers for their positive comments and suggestions to improve our manuscript. All alterations included in the manuscript are highlighted in blue.
REVIEWER 02
- It will be interesting to provide further info for why p.R337His high in other region but low in midwestern region? Any population history info, dating info etc. to enhance the explanation?
Response: The TP53 p.R337H variant and its impact on cancer development has been studied in our country by several authors from the South and Southeast Brazilian regions. We summarized the literature data on page 6, lines 212 – 221.
- There are multiple grammatical errors in table and poor quality of figures that Figure 1 is largely messed up for patient information Line 136 – The text written 161 out of 217 are female, while in table 1, 149 out of 217 are females.
Response: The manuscript was revised and proofread, as recommended. Our sincere apologies for the mistakes found in the table and the figure. The figure was now modified and presented in high resolution.
- Inconsistent reference format with the main manuscript.
Response: The reference section followed the Journal recommendations and was revised accordingly.

Round 2
Reviewer 2 Report
The revision has satisfactorily addressed the questions I had for the original version, and improve the quality.